# Prescription of Choreito, a Japanese Kampo Medicine, with Antimicrobials for Treatment of Acute Cystitis: A Retrospective Cohort Study

**DOI:** 10.3390/antibiotics11121840

**Published:** 2022-12-18

**Authors:** Toru Sugihara, Jun Kamei, Hideo Yasunaga, Yusuke Sasabuchi, Tetsuya Fujimura

**Affiliations:** 1Department of Urology, Jichi Medical University, Tochigi 329-0498, Japan; 2Department of Clinical Epidemiology and Health Economics, School of Public Health, The University of Tokyo, Tokyo 113-8654, Japan; 3Data Science Center, Jichi Medical University, Tochigi 329-0498, Japan

**Keywords:** acute cystitis, administrative database, Choreito, female, Japan, Japanese herbal medicine, JMDC claims database, Kampo medicine

## Abstract

Choreito, a Japanese Kampo medicine, is used to treat Japanese female patients for the quick relief of inflammatory symptoms associated with acute cystitis. We evaluated whether Choreito is effective in reducing antibiotic use and the number of clinic visits for these patients. Females aged 18–49 years who had acute cystitis for the first time, with no history of medical insurance use within 90 days prior to their visit, and no hospitalizations within the 30 days after their first visit were identified from the JMDC Claims Database between April 2018 and March 2021. For the 30 days after their first visit, patients who were given their first antimicrobial prescriptions with or without Choreito were compared regarding (i) the number of clinic visits, (ii) total antimicrobial prescription days, and (iii) the number of antimicrobial prescriptions adjusted for their age, Charlson comorbidity index, and the COVID-19 pandemic period (after April 2020). For the 319 and 8515 patients with or without a Choreito prescription, respectively, multivariable Poisson regression analyses showed that Choreito was significantly associated with a 5% shortening of a patient’s total antimicrobial prescription days (Beta, 0.950; *p* = 0.038), whereas no significant difference was observed in the number of clinic visits and antimicrobial prescriptions (*p* = 0.624 and *p* = 0.732, respectively). The prescription of Choreito in combination with antimicrobials was associated with a slight reduction in total antimicrobial use for acute cystitis among females.

## 1. Introduction

Acute cystitis is the most frequent infection encountered in general clinical practice; approximately one in three women have been reported to experience urinary tract infections at least once in their lifetime [1,2,3]. Antibiotics are often used to treat acute cystitis. The effect of antibiotics in urinary tract infections is an important topic in today’s world; however, excessive antibiotic use can be a problem as it promotes microbial systemization and makes the treatment of infections more difficult. The high cost of antibiotics is also a concern and has become a social problem; countermeasures against this problem are urgently needed [4,5,6,7,8].

Choreito is a traditional Japanese Kampo medicine that is often used as a treatment for acute cystitis in females. Kampo medicine derives from traditional Chinese medicine that was adopted in Japan in the sixth century, and the prescription of some Kampo medicines is covered by the Japanese national health insurance system [9].

Choreito is a medicinal herb composed of five aqueous extracts, including aluminum silicate hydrate with silicon dioxide, *Alisma* rhizome (a rhizome of *Alisma orientale* Juzepczuk), *Polyporus umbellatus* sclerotium, *Poria* sclerotium (a dried sclerotium of *Wolfiporia cocos* Ryvarden et Gilbertson), and Donkey glue. Several studies have reported supportive data regarding the effects of Choreito on the alleviation of lower urinary tract symptoms in patients who complain of decreased urine volume, dysuria, and oral dryness [10,11,12,13,14,15,16]. In the context of Kampo medicine, Choreito is clinically considered to “dampen heat” in the lower abdomen, characteristic symptoms of which include dysuria, heat in the lower abdomen, and thirst [17]. Among Kampo medicines, Goreisan and Seishinrenshiin are also considered to be effective for symptom relief in acute cystitis; however, they are mainly effective for relieving frequent urination and dysuria, and are not thought to have much of an effect on inflammation due to cystitis, unlike Choreito. Based on these data, Choreito is frequently used to treat urological diseases, such as cystitis, lower urinary tract symptoms, and an overactive bladder. Although most Japanese physicians base their practice on Western medicine, they often use Kampo preparations in combination with Western medicine. It is not unusual for Japanese physicians to prescribe both modern Western medicines and traditional Kampo herbal preparations for the same patient [18,19,20].

However, limited information is available on the clinical benefits of Choreito for acute cystitis. Although the effectiveness of Choreito for severe hemorrhagic acute cystitis caused by chemotherapy in pediatric patients with specific blood diseases has been suggested [17,21], to the best of our knowledge, there is no English literature that presents clinical data on the use of Choreito for treating simple acute cystitis. We hypothesized that if Choreito has the clinical benefit of improving the symptoms of cystitis, it would decrease the number of clinic visits and the amount of antimicrobial use among patients with acute cystitis.

The present study aimed to evaluate the effects of Choreito for female patients treated with antimicrobials for acute cystitis based on a large administrative claims database in Japan.

## 2. Results

A total of 29,043 females aged 18–49 years had experienced acute cystitis for the first time. Among them, 13,625 patients had no history of medical insurance use within the 90 days prior to their first clinic visit for acute cystitis. Therefore, 15,418 patients were excluded. After excluding those who had been hospitalized within 30 days after their first visit for acute cystitis (13 patients), 13,612 females remained. After excluding patients whose prescription at their first visit contained medicines other than antimicrobials or Choreito (4778 patients), 319 and 8515 female patients with acute cystitis with or without a prescription for Choreito, respectively, were identified. A detailed patient selection flowchart is shown in Figure 1. 

The baseline characteristics and outcomes of the two groups are described in Table 1. No significant differences in the patients’ backgrounds were observed, except for age.

Of the three endpoints, the total number of antibacterial prescription days was the only outcome that exhibited significant differences between the two groups. The average of the total number of antibacterial prescription days for treatment with or without Choreito was 5.41 and 5.67 days, respectively. This difference was less significant.

The multivariable Poisson regression analysis in Table 2 revealed that the prescription of Choreito was significantly associated with a 5% shortening of patients’ total antibacterial prescription days (Beta, 0.950, *p* = 0.038). However, for the number of clinic visits within 30 days and the number of antibacterial prescriptions within 30 days, the difference between the two groups did not reach a level of significance (*p* = 0.624 and *p* = 0.732, respectively).

Advancing age and the date after the state of emergency was declared were significantly associated with an increase in the number of visits (+12.7%, *p* < 0.001 and +8.5%, *p* = 0.011, respectively) and total antimicrobial days (+1.5%, *p* = 0.001 and +1.8%, *p* = 0.049, respectively). 

## 3. Discussion

The present study, using a Japanese nationwide administrative claims database, revealed that Choreito, a Japanese traditional Kampo medicine, could reduce antimicrobial use by 5%. However, Choreito did not significantly impact the number of clinic visits and antibacterial prescriptions within 30 days.

The abuse of antimicrobials has a negative public health impact. For 70 years, harmful infections have been treated by antimicrobials; however, the development of resistance mechanisms by microorganisms against antimicrobials has hampered therapeutic efficacy, and no effective alternative is currently available [4,5,7].

Economic impacts are also expected, as many countries and pharmaceutical companies have policies to increase funding for antimicrobial-related medical and drug development [6].

Although the widespread application of antimicrobials will reduce microbial contamination in the treatment of diseases, there is concern that the release of large quantities of antimicrobials into the environment will accelerate the emergence of new microorganisms that would be resistant to these antimicrobials. If the number of antimicrobial-resistance-acquired microorganisms continues to increase, the infectious diseases caused by these microorganisms would be difficult to treat and will present a hidden danger to public health for decades to come [8,22]. Therefore, attempts to reduce antibiotic use are important. As urinary tract infections are very frequent and affect one out of three women in their lifetime, antibiotics are frequently administered, and attempts to reduce the dosage of antibiotics for acute cystitis are highly significant [1,2,3].

Kampo extracts are crude drugs derived from minerals, plants, and animals, and their quality is controlled through quantitative analysis of the sample ingredients using high-performance liquid chromatography. In the United States, they are approved for sale by the Food and Drug Administration [17]. In Japan, more than 230 types of Kampo medicines have already been officially approved as pharmaceutical products by the Japanese Ministry of Health, Labour, and Welfare [9,23,24,25]. They are routinely used for the treatment of a wide range of diseases. Many Kampo medicines are manufactured on a modern industrial scale, and the quality and quantity of ingredients are standardized under scientific quality control. In the field of urology, some Kampo medicines, including Choreito, are used to treat urological diseases, such as cystitis, lower urinary tract symptoms, and an overactive bladder [18,19,20].

As mentioned above, in Japanese clinical practice, Kampo is widley accepted in daily clinical practice [23,26], and Choreito is widely prescribed for the relief of lower urinary tract symptoms, especially for dysuria, decreased urine output, and dryness of the oral cavity [16,27]. Some constituents of Choreito, such as aluminum silicate hydrate, *Alisma* rhizome, *Polyporus umbellatus* sclerotium, and *Poria* sclerotium, are said to have diuretic effects and improve water imbalance in the body [10]; however, the biological response of each of these components is not fully understood.

Various other health-improving biological responses have been reported for the constituents of Choreito. Aluminum silicate hydrate has been reported to exert anti-tumor effects in Wistar rats [14]. *Alisma* rhizome has been reported to lower urinary pH and inhibit the formation of struvite crystals in urine [11]. Aqueous extracts of *Polyporus umbellatus* sclerotium are known to be effective for the prevention of early renal damage in rat models of nephropathy [13]. *Poria* sclerotium has data demonstrating its efficacy against proto-glomerular basement membrane nephritis in rats through its main element, pachyman [15]. It has been suggested that it exerts anti-aging effects by enhancing antioxidant activity, scavenging free radicals, and modulating the expression of aging-related genes [12].

Despite these basic reports, the evidence supporting the efficacy of Choreito in clinical medicine is limited. Only two reports have been found in the English literature regarding the effect of Choreito on acute cystitis. Kawashima et al. [17,21] reported that Choreito helped to treat bladder hemorrhage in a pediatric patient with refractory acute lymphoblastic leukemia who developed uncontrolled gross hematuria. They also reported that Choreito was effective in the treatment of BK-virus-associated hemorrhagic cystitis. Six patients treated with Choreito reported a significantly shorter time to reach complete remission for hemorrhagic cystitis compared to eight patients who were not treated with Choreito (9 days vs. 17 days, *p* = 0.037).

To overcome these limitations, we examined the efficacy of Choreito for the treatment of common acute simple cystitis. The results showed no difference in the number of clinic visits or the number of antimicrobial prescriptions issued. Although a 5% decrease in antimicrobial use was observed, this difference was not likely to be clinically significant. This is because, generally, Kampo herbal medicine has milder effects than Western medicine [28].

We speculate that a slight decrease in antibacterial use could be due to the combination of Choreito and antibacterial use, which may have been able to achieve early improvements for bleeding and dysuria caused by cystitis, leading to faster amelioration of patients’ subjective symptoms and reducing their need for an additional antibacterial prescription. However, it is difficult to demonstrate this from the present database study.

Another notable finding from the multivariate analyses was that the number of days of antibacterial prescriptions for acute cystitis and the number of hospital visits increased slightly (by 1.8% and 4.1%, respectively) after the coronavirus disease-19 (COVID-19) pandemic. This could have been because, during the pandemic, there was great social turmoil, such that the Tokyo Olympics in summer was postponed as a measure of COVID-19 control. Therefore, patients with high fever were at a great disadvantage because they could not distinguish fever from COVID-19 infection. Hence, patients were administered antimicrobials for a longer period of time in order to avoid severe urinary tract infections, such as pyelonephritis. However, this could not be proven in the present study. The review papers analyzing the COVID-19 pandemic outbreak and antibiotic administration in non-COVID-19-infected patients reported mixed results of decreased or increased antibiotic use; therefore, definitive conclusions could not be drawn [29,30,31,32].

It is known that some patients with COVID-19 have lower urinary tract symptoms, which are referred to as COVID-19-associated cystitis [33]. The severity and frequency of these symptoms remains unclear. The number of COVID19-infected people in Japan as of March 2021 was much smaller than that in Europe and the United States. The total number of patients was about 450,000 (0.3% of the total population). Therefore, our understandings of the impact of COVID-19-associated cystitis on the present study is limited.

Similarly, the frequency of revisits and the duration of antimicrobial therapy tended to be higher in cases of increased age, which could be due to the higher probability of antibiotic resistance with increasing age due to a history of antibacterial use. However, the exact cause remains unknown. It could simply be that younger patients are less likely to return to the hospital for a second visit due to a lack of time and, as a result, are less likely to receive a second or subsequent antibacterial prescription. The observed differences were so slight that it was difficult to identify a definitive reason.

The present study had several limitations. The main limitation of our study is that it was based on retrospective administrative claims data. The diagnoses of acute cystitis were based on the judgment of doctors; therefore, over- or under-estimation cannot be completely ruled out. From the present administrative claims database, it is not possible to determine whether the diagnosis of acute cystitis was based solely on the patient’s complaint, urinalysis, or the extent to which urine culture was used as a reference. Similarly, it is difficult to ascertain how the physician decided that the patient was cured. Second, we could not evaluate detailed clinical variables, such as vital signs, laboratory and bacterial culture results, and imaging findings. Third, Choreito can be purchased over the counter outside of medical insurance, which would not have been captured in this database. Fourth, this study was not designed to prospectively randomize patients. Therefore, we cannot rule out the possibility that the prescription of antibacterials was lower among physicians who administered Choreito because they tended to have a better understanding of drug-resistant bacteria and were more concerned about the proper use of antibiotics. Similarly, there may be other hidden confounding factors that were not visible in the present database. Patients with more severe pain could have been more likely to receive Choreito. Therefore, the results might have been biased due to this unmeasured confounder. However, its effect probably works in the direction of reducing the efficacy of Choreito, so it can be said that the present results are more robust. Fifth, the patient selection process in the present study excluded hospitalized cases or women who visited multiple medical facilities 30 days after their first visit for acute cystitis. This could be a bias as hospitalized cases with severe drug-related side effects or serious conditions, such as pyelonephritis, due to worsening urinary tract infections, might have been excluded. However, we believe that the impact of this is limited due to the safety aspect of Choreito and the fact that only 13 cases were excluded due to hospitalization.

Despite these limitations, the present research, in which a large number of cases were compared, could contribute to building a certain amount of reliable evidence.

## 4. Materials and Methods

### 4.1. Data Source 

This retrospective cohort study was conducted using the JMDC Claims Database (Tokyo, Japan) between April 2018 and March 2021. It is a commercially available database, accessible to everyone once purchased from JMDC Inc., Tokyo, Japan [34,35]. (https://www.jmdc.co.jp/en/jmdc-claims-database/, accessed on 18 December 2022) The company is a commercial enterprise that charges a fee for updating, maintaining, managing, and providing access to the database. The JMDC claims database contains reimbursement data for in- and out-patient insurances. In 2022, it registered more than 14 million individuals. The database includes administrative claims records registered from more than 60 healthcare insurers in Japan and tends to register people working for large companies, most of whom are under 65 years of age; no one over 75 years is registered [36].

The claims data include several clinical pieces of information, such as (i) diagnoses based on the International Classification of Diseases, 10th Revision (ICD 10) codes, and their initial and end dates; (ii) the dates of laboratory and medical imaging tests, surgeries, and medical procedures; (iii) admission and discharge times; and (iv) details of prescriptions. 

### 4.2. Ethical Issue

This study was performed under the ethical guidelines of the University of Tokyo (approval by the Ethical Committee of the University of Tokyo: 2018 10862) and the principles of the Declaration of Helsinki. Data in the JMDC Claims Database were deidentified, and informed consent was not required.

### 4.3. Study Population

We extracted outpatients’ data between April 2018 and March 2021. In the present study, we aimed to limit our analysis to female patients experiencing cystitis for the first time who had few underlying medical conditions and did not require regular clinical visits. Therefore, selected cases met the following criteria: (i) females aged 18 to 49 years who suffered from acute cystitis (ICD-10, N300) for the first time and had visited only one clinic within 30 days of their first visit; (ii) no history of medical insurance use within 90 days prior to their first visit for acute cystitis; (iii) no hospitalization within 30 days after their first visit for acute cystitis; (iv) a prescription at their first visit that included only antimicrobials or only antimicrobials in combination with Choreito. The first visit for acute cystitis was defined by the date of registration of the Japanese healthcare insurance code A000 (first visit to a medical institution). The second and subsequent visits to the same clinic were determined by the date on which A001 or A002 (second visit to a clinic) occurred.

Data including age, sex, and comorbidities at their first visit for acute cystitis were also extracted. Any comorbidities at admission were converted into Charlson comorbidity index (CCI) scores, according to the protocol of Quan et al. [18]. The higher the score, the more likely it was that the predicted outcome would result in mortality or higher resource use. The patients were divided into two groups: treatment with antimicrobials with or without Choreito.

The COVID-19 pandemic was also added as a variable as it is a time factor that cannot be ignored. In Japan in April 2020, the number of people infected with the new coronavirus increased rapidly, especially in urban areas such as Tokyo and Osaka. The government declared a state of emergency for the first time in seven prefectures, considering the high risk of an overwhelmed and dysfunctional medical care delivery system. The first statement due to the COVID-19 pandemic was issued during the observation period of this study, after which daily activities, including medical care, changed significantly. Therefore, in the multivariable analysis, we also added a factor of whether acute cystitis occurred before or after a state of emergency was declared.

### 4.4. Endpoints

Three endpoints were evaluated: (i) the number of clinic visits within 30 days of the first visit for acute cystitis, (ii) the total number of antibacterial prescription days, and (iii) the number of antibacterial prescriptions within 30 days.

### 4.5. Statistical Analysis

Patient characteristics were compared using the Mann–Whitney U test and chi-squared tests. Multivariable Poisson regression analysis for the three endpoints was performed with variables including age, CCI, the date of onset after or before the first statement of emergency for COVID-19, and treatment with or without Choreito. All statistical analyses were performed using SPSS version 25.0 (IBM Corp., Armonk, NY, USA) [20]. Statistical significance was defined as a *p*-value of <0.05. 

## 5. Conclusions

The prescription of Choreito in combination with antimicrobials is associated with a slight reduction in total antimicrobial use among females with acute cystitis. The emergence of antibiotic-resistant bacteria, due to the extensive use of antibiotics, has become a public health problem, and Choreito could be a tool that contributes to curbing the use of antibiotics, even if only slightly.

## Figures and Tables

**Figure 1 antibiotics-11-01840-f001:**
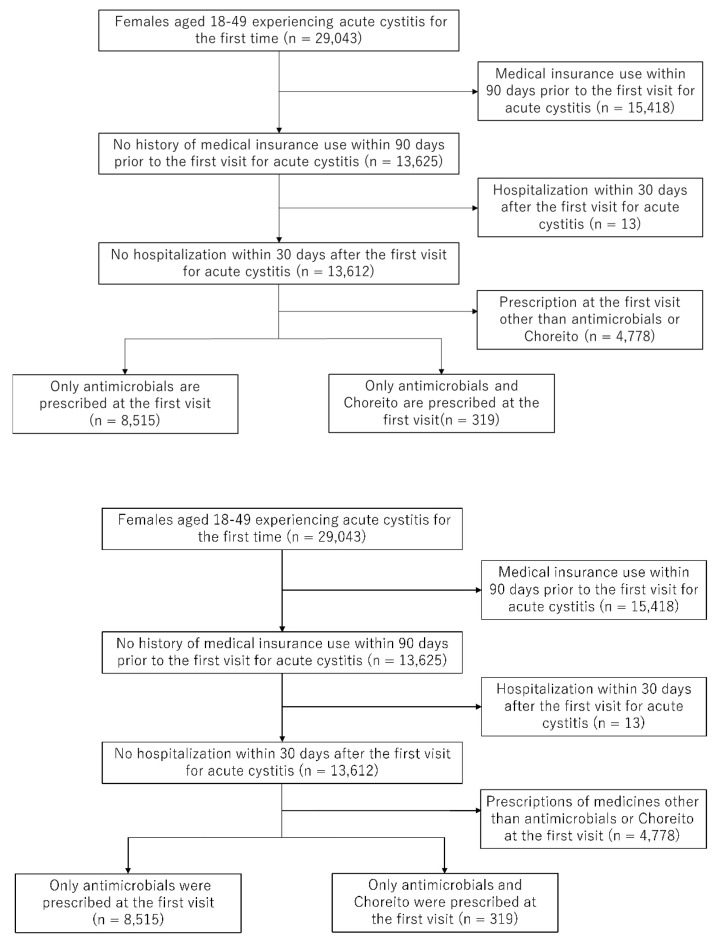
Flow chart outlining patient selection.

**Table 1 antibiotics-11-01840-t001:** Characteristics and outcomes of 8834 females with acute cystitis receiving antimicrobials with or without Choreito (Kampo medicine) at their first clinic visit between April 2018 and March 2021.

	Prescription for Acute Cystitis n (%), or Median (IQR)		*p*-Value
	Antimicrobials n = 8515	Antimicrobials with Choreito n = 319	
Female	8515 (100)	319 (100)	–
Age	32 (24–42)	34 (26–43)	0.014
Charlson comorbidity index			
0	6321 (74.2)	229 (71.8)	0.543
1	1776 (20.9)	71 (22.3)	
≥2	418 (4.9)	19 (6.0)	
The state of emergency against COVID-19 in Japan			
Before (2018 April to 2020 March)	5126 (60.2)	179 (56.1)	0.143
After (2020 April to 2021 March)	3389 (39.8)	140 (43.9)	
OUTCOME			
The number of clinic visits within 30 days	1 (1–2)	1 (1–2)	0.885
1	5169 (60.7)	194 (60.8)	0.549
2	2898 (34.0)	104 (32.6)	
≥3	448 (5.3)	21 (6.6)	
Total antibiotic prescription days	5 (5–7)	5 (4–7)	0.002
1 to 3 days	1021 (12.0)	54 (16.9)	0.041
4 to 5 days	4587 (53.9)	170 (53.3)	
6 to 7 days	2163 (25.4)	68 (21.3)	
≥8 days	744 (8.7)	27 (8.5)	
The number of antibiotic prescriptions within 30 days	1 (1–1)	1 (1–1)	0.476
1	7667 (90.0)	291 (91.2)	0.488
≥2	848 (10.0)	28 (8.8)	

Data are presented as number (percentage) or median (interquartile range). COVID-19, coronavirus disease 2019.

**Table 2 antibiotics-11-01840-t002:** Multivariable Poisson regression analyses of 8834 patients who were prescribed antimicrobials for acute cystitis with or without Choreito (Kampo medicine).

Factors	Beta (95% CI)	*p* Value
Endpoint 1. The number of clinic visits within 30 days		
Age, by 10 years	1.127 (1.091 to 1.164)	<0.001
Charlson comorbidity index (continuous)	1.039 (0.995 to 1.085)	0.085
After the state of emergency against COVID-19 in Japan (vs. before)	1.085 (1.019 to 1.155)	0.011
Choreito prescription (yes vs. no)	1.041 (0.887 to 1.222)	0.624
Endpoint 2. Total antibiotic prescription days		
Age, by 10 years	1.015 (1.006 to 1.025)	0.001
Charlson comorbidity index (continuous)	0.998 (0.984 to 1.011)	0.713
After the state of emergency against COVID-19 in Japan (vs. before)	1.018 (1.000 to 1.036)	0.049
Choreito prescription (yes vs. no)	0.950 (0.906 to 0.997)	0.038
Endpoint 3. The number of antibiotic prescriptions within 30 days		
Age, by 10 years	1.010 (0.990 to 1.032)	0.326
Charlson comorbidity index (continuous)	1.002 (0.972 to 1.032)	0.905
After the state of emergency against COVID-19 in Japan (vs. before)	1.008 (0.968 to 1.049)	0.712
Choreito prescription (yes vs. no)	0.982 (0.882 to 1.092)	0.732

CI, confidential interval; COVID-19, coronavirus disease 2019.

## Data Availability

The JMDC database is a paid database that anyone can access after access is purchased from JMDC Inc., Tokyo, Japan (https://www.jmdc.co.jp/en/jmdc-claims-database/, accessed 18 Decembre 2022).

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
