# Peer review of "Prescription of Choreito, a Japanese Kampo Medicine, with Antimicrobials for Treatment of Acute Cystitis: A Retrospective Cohort Study"

_antibiotics, 2022, doi:10.3390/antibiotics11121840_

Round 1

Reviewer 1 Report

It is well known that because of emerging antibiotic resistance any alternative should be considered. This research is important because it tests the effect of popular remedies used under the name Choreito to complete treatment of acute cystitis in order to reduce the dosage and period of antibiotic use.

The topic is relevant in the field even if the results show a relatively small efficiency but they are performed on a large enough cohort to be relevant.

The methods are clearly and logically formulated, the flow chart of patient selection is easy to follow, the results are well presented in tables cited in the text of the manuscript, the use of references is adequate without self-citations, discussions and conclusions are described in a scientific manner, easy to understand, in accordance with the subject of the manuscript.

English formulation is adequate, statistical analysis adds to the value of the study.

My opinion on this manuscript is to be accepted as it is.

Author Response

We thank you very much for your detailed review of our paper.

Reviewer 2 Report

The topic is good and will contribute to the research and be beneficial to the scientific community. However, the article needs to be written in-depth, and would be better to consider the below comments for improvements and mention the relevant things as per the title of the article:

Needs to check the format and subtitle of the article as per journal guidelines.

1.    Abstract:

The abstract of 208 words is provided – it would be better if abstracts reflect the objective very clearly.

2.    Introduction:

Needs to mention the details of Choreito and which type of Kampoo is considered for this study and why?

3.    Results:

Table 1: Comparing the Antimicrobials Vs Antimicrobials with Choreito there is only a slight difference in the results.  Also, the number of participants is not the same – you can give the rationale for comparison and how reliable to get a conclusion from this – Maybe add on as a limitation of the study?

You can add on the part - How COVID-19 affected the study conduct [Long COVID and COVID-19-associated cystitis (CAC)]?

4.    Materials and Methods:

This part is clear and understandable.  

5.    Conclusions

The conclusion is written well.

6.    References:

All 35 references are cited in the text and mentioned in the paper.   

Reviewer 3 Report

Attached
